# Distinct stages of the translation elongation cycle revealed by sequencing ribosome-protected mRNA fragments

**Liana F Lareau*[†‡], Dustin H Hite[†], Gregory J Hogan, Patrick O Brown**

Department of Biochemistry, Howard Hughes Medical Institute, Stanford University School of Medicine, Stanford, United States

**Abstract** During translation elongation, the ribosome ratchets along its mRNA template, incorporating each new amino acid and translocating from one codon to the next. The elongation cycle requires dramatic structural rearrangements of the ribosome. We show here that deep sequencing of ribosome-protected mRNA fragments reveals not only the position of each ribosome but also, unexpectedly, its particular stage of the elongation cycle. Sequencing reveals two distinct populations of ribosome footprints, 28–30 nucleotides and 20–22 nucleotides long, representing translating ribosomes in distinct states, differentially stabilized by specific elongation inhibitors. We find that the balance of small and large footprints varies by codon and is correlated with translation speed. The ability to visualize conformational changes in the ribosome during elongation, at single-codon resolution, provides a new way to study the detailed kinetics of translation and a new probe with which to identify the factors that affect each step in the elongation cycle.

*For correspondence: lareau@berkeley.edu

[†]These authors contributed equally to this work

**Present address:** [‡]California Institute for Quantitative Biosciences, University of California, Berkeley, United States

**Competing interests:** The authors declare that no competing interests exist.

**Reviewing editor**: Roy Parker, University of Colorado, United States

## Introduction

To accomplish the huge task of translation elongation—in each cycle, accurately incorporating a new amino acid into a nascent peptide every 1/6[th] of a second, then moving precisely three nucleotides along the mRNA template—the ribosome undergoes a series of major structural rearrangements (*Figure 1*) (reviewed in *Chen et al., 2012* and *Noeske and Cate, 2012*). During the initial decoding step of elongation, aminoacylated tRNAs are delivered to the decoding site (A site) as part of a ternary complex with EF-Tu (in prokaryotes) or the orthologous eEF1A (in eukaryotes). When the anticodon of one of these aminoacylated tRNAs is able to base-pair stably with the specific mRNA codon in the decoding site (A site), a new peptide bond is formed between the nascent polypeptide and the specified amino acid. The ribosome then undergoes a massive rearrangement in which the ribosomal subunits rotate relative to each other (*Frank and Agrawal, 2000*; *Zhang et al., 2009*). Along with this rotation, the A and P site tRNAs move from 'classic' to 'hybrid' states: the anticodon ends stay in their original A and P sites and the acceptor ends move to the P and E sites (*Moazed and Noller, 1989*; *Munro et al., 2007*). This rotated state of the ribosome undergoes additional conformational changes in preparation for translocation (*Zhang et al., 2009*; *Fu et al., 2011*). The ribosome can fluctuate between rotated and non-rotated states until EF-G (eEF2 in eukaryotes) binds and stabilizes the rotated ribosome (*Agirrezabala et al., 2008*). GTP hydrolysis by EF-G then promotes translocation of the mRNA along the ribosome, coupled to a large intra-subunit rotation of the 30S head (*Ratje et al., 2010*), after which the ribosome subunits rotate back to a closed formation for the next cycle (*Gao et al., 2009*). Structural and biochemical studies have revealed many of the atomic-level changes that allow this complicated process to occur (*Pulk and Cate, 2013*; *Tourigny et al., 2013*; *Zhou et al., 2013*), and new details continue to emerge, reshaping models, raising new questions, and leaving other questions still unanswered.

**eLife digest** To make a protein from a gene, the gene is first transcribed to produce a molecule of messenger RNA (mRNA), which then passes through a molecular machine called a ribosome. The ribosome reads the genetic code in the mRNA in groups of three letters at a time, and each triplet of letters (or codon) represents an amino acid. The ribosome then joins the relevant amino acids together to build a protein.

The ribosome processes about six amino acids per second, on average, but the mRNA is not fed through at a constant rate. Instead, the ribosome changes its shape to ratchet along the mRNA from one codon to the next: it then reads the new codon and adds another amino acid to the protein. However, many of the details of this ratcheting process are not fully understood.

In this study, Lareau, Hite et al. have used a technique called 'ribosome profiling' to explore the movement of ribosomes along mRNA molecules. First, all of the pieces of mRNA molecules that are not protected inside a ribosome were chemically destroyed. The sequences of the protected fragments were then read and matched to the full-length gene sequences. The protected fragments came in two different sizes: some were about 28–30 letters long, and others were about 20–22 letters long. Lareau, Hite et al. suggest that these different fragment sizes occur because the ribosome switches between two shapes at each codon as it ratchets along the mRNA, and so it protects different lengths of mRNA.

In previous ribosome-profiling experiments, the fragments had all been about 28 letters long; but these experiments had used a chemical to halt the progress of the ribosomes along the mRNAs before measuring the length of the fragments. Lareau, Hite et al. argue that this chemical locks the ribosome in the same shape when it brings the ribosome to a halt, and so the protected fragments always have the same length. Further, other chemicals that halt ribosomes appear to lock this molecular machine in the other shape, and so it can only protect the shorter fragments.

The findings of Lareau, Hite et al. show that ribosomal profiling experiments can reveal much more than simply where a ribosome is on an mRNA molecule. Further study into the different stages of the ribosome ratcheting process will help uncover how the speed that a ribosome translates an mRNA into a protein can be encoded in the mRNA sequence itself.

Recently, 'ribosome profiling' by high-throughput sequencing of ribosome-protected fragments has provided a powerful tool for identifying the position of ribosomes on mRNAs across the entire transcriptome (*Ingolia et al., 2009*). Cell lysates are treated with nuclease to degrade all mRNA not physically protected by ribosomes, and the ribosome-protected fragments are extracted, sequenced, and mapped back to the genome to show ribosome positions, revealing the overall translation level of each gene as well as the distribution of ribosomes along the mRNA. Nucleotide-level precision of ribosome positions is possible because of the very consistent size of ribosome footprints in the conditions assayed. The authors of the method used a nuclease protection assay to establish that, in yeast treated with the elongation inhibitor cycloheximide, each ribosome protects a footprint of 28 nucleotides (nt), confirming earlier reports (*Steitz, 1969*; *Wolin and Walter, 1988*).

While performing ribosome-profiling experiments in *Saccharomyces cerevisiae*, we serendipitously noticed a population of smaller ribosome-protected fragments. To better capture these fragments and to investigate their origins, we revised the ribosome-profiling protocol originally established by Ingolia et al. Our experiments revealed that, in the absence of cycloheximide, the small ribosome-protected fragments were abundant, consistent with an early observation of short ribosome footprints in the absence of cycloheximide (*Wolin and Walter, 1988*). We show here that the small fragments originate from ribosomes in a conformation distinct from that previously observed in the presence of cycloheximide. The ability to discern distinct ribosomal structural states by ribosome profiling has given us insight into how codon, tRNA, and amino acid identity and translational speed relate to ribosome structure. This additional dimension of ribosome-profiling data will provide a valuable new layer of molecular and mechanistic information, at codon resolution, for future studies of translation.

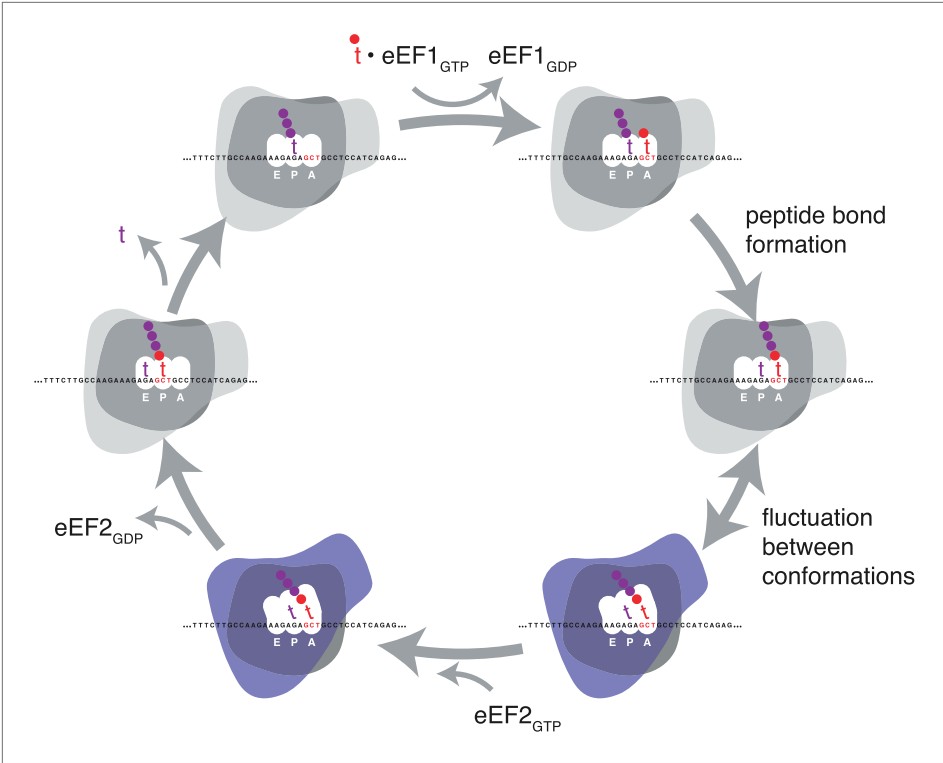

**Figure 1**. Schematic representation of the eukaryotic elongation cycle. Blue overlay denotes stages at which the ribosome has undergone a large inter-subunit rotation. Ribosome shapes are for illustration only, not a literal representation of the structure or degree of rotation.

## Results

### Ribosomes can protect two distinct mRNA fragment sizes

We began our investigation of ribosome footprint size by isolating ribosome-protected mRNA fragments from yeast using a modified ribosome-profiling procedure. The standard ribosome-profiling protocol includes a size selection for RNA fragments of around 28 nt. To eliminate the bias against smaller fragments, we broadened the initial size range and selected RNA fragments between 18 and 32 nt after RNase I digestion. By selecting fragments in this broader size range, and, importantly, by carrying out the entire procedure in the absence of cycloheximide or other inhibitors, we observed two clearly distinct, abundant populations of ribosome-protected mRNA fragments ('footprints'), 28–30 nt and 20–22 nt long. We visualized fragment lengths and positions with a three-dimensional 'metagene' representation: sequence reads representing the ribosome-protected fragments from all expressed genes were aligned relative to the start codon of the corresponding gene and tallied by fragment length and position to show the average pattern of translation along all annotated coding regions (*Figure 2A–C*, *Figure 2—figure supplement 1*).

We found overwhelming evidence that both populations of fragments came from translating ribosomes. The 21 and 28 nt fragments were both found almost entirely within annotated coding regions (CDS) and not in 5' or 3' UTRs; 98.3–99.7% of mappable 21 nt fragments, and 96.5–99.6% of mappable 28 nt fragments, mapped within the annotated CDS in three replicates (*Figure 2D*). Both populations also showed the 3-nucleotide periodicity expected of fragments originating from elongating ribosomes (*Figure 2E*). We conclude that fragments of both sizes are footprints of translating ribosomes.

The 5'-most peaks in the metagene represent ribosomes with the start codon in the P site and the second codon in the A site (*Kapp and Lorsch, 2004*; *Ingolia et al., 2009*). Using this as a reference for phasing all the footprints, we inferred that for ribosomes with a given codon in the A site, small and large footprints generally had the same 5' ends positioned 15–16 nt upstream of the A-site codon, and

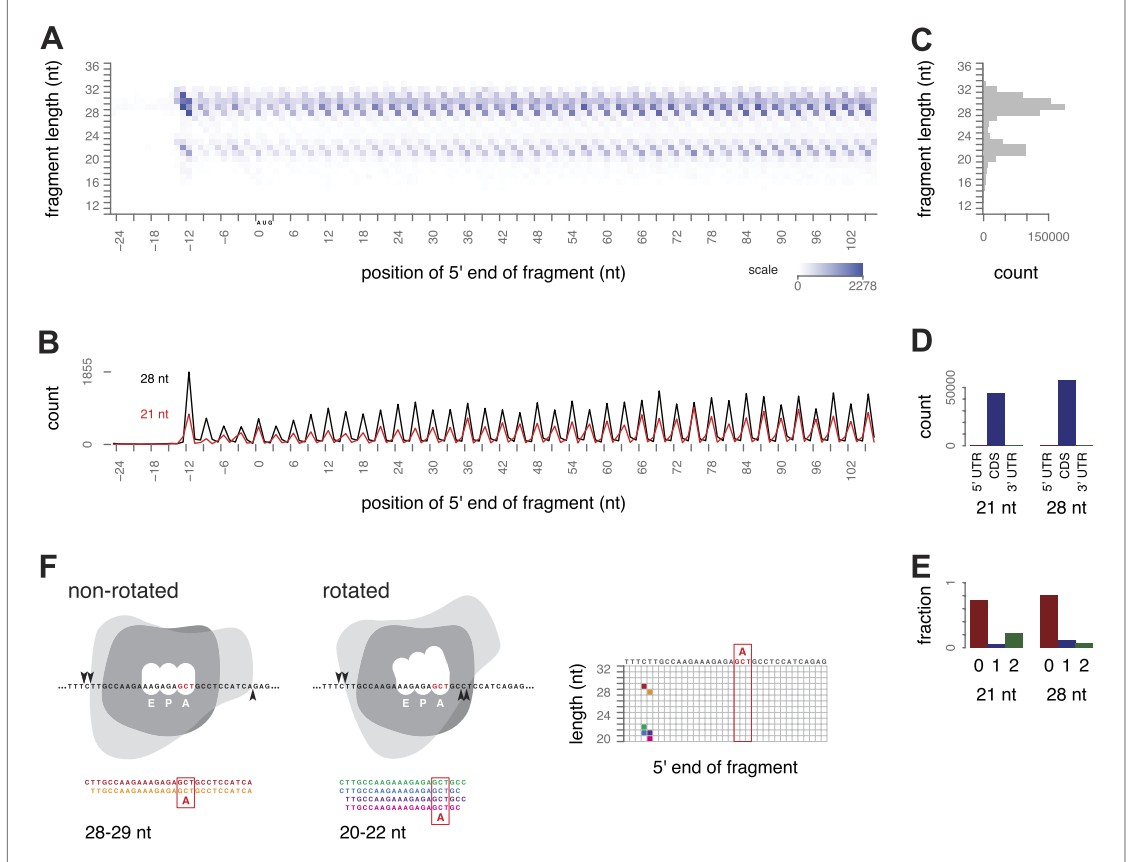

**Figure 2**. Ribosome-protected fragment positions and size distributions from yeast not treated with elongation inhibitors. (**A**) The position of each fragment was calculated relative to the start codon of its gene. The 5′ end positions (x axis) and lengths of all fragments (y axis) were tallied across all genes with a coding region of at least 300 nt. Higher color intensity reflects more fragments. RNA fragments between 18 and 32 nucleotides were selected after gel electrophoresis; shorter and longer fragments are not entirely excluded but their read counts are presumed to be unrepresentative of their true abundance. (**B**) Profiles of the 5′ end positions of all 20 nt and 28 nt fragments relative to the start codon of their genes, as in (**A**). (**C**) Total counts of mapped fragment lengths. (**D**) Distribution of 21 nt and 28 nt fragments in coding regions and untranslated regions of mRNAs. (**E**) Positions of 21 nt and 28 nt fragments relative to the reading frame. (**F**) Interpretation of fragment positions on an arbitrary gene fragment. Arrowheads show hypothetical nuclease cleavage sites relative to a ribosome in a non-rotated or rotated conformation (shape is for illustration only). The resulting fragments are shown with the inferred decoding site (A site), and their positions in a grid as in *Figure 2A* are shown with corresponding colors.

The following figure supplements are available for figure 2:

**Figure supplement 1**. Ribosome-protected fragment positions and size distributions from yeast not treated with elongation inhibitors.

differed at their 3′ ends: extending 2–3 nt beyond the A-site codon in the small footprints and 10 nt beyond the A-site codon in the large footprints, respectively (*Figure 2F*).

## Different elongation inhibitors stabilize distinct conformations and bias the footprint size distribution

During elongation, at each codon, the ribosome cycles through a stereotyped sequence of steps as it incorporates the specified amino acid and translocates to the next codon. These steps are accompanied by major rearrangements of the ribosome structure, including a rotation of the large subunit relative to the small subunit upon peptide bond formation. We hypothesized that the non-rotated, pre-peptide-bond ribosomes and rotated, post-peptide-bond ribosomes might protect different lengths of mRNA, and that the two resulting footprint sizes might, therefore, represent these two conformations.

To determine what footprint sizes were protected by ribosomes in distinct stages of elongation, we performed ribosome profiling on yeast treated with inhibitors that block different steps of the cycle.

Cycloheximide is an elongation inhibitor that binds to the E site of ribosomes, preventing the E site tRNA from leaving the ribosome. When cycloheximide was added to the yeast immediately before harvest and was present throughout lysis and RNase I treatment, the most prevalent footprints were 28–30 nt long and were distributed along the coding sequence with a 3-nt periodicity (*Figure 3A–C*, *Figure 3—figure supplement 1*). Apart from a distinct peak at the start codon, there were very few 20–22 nt footprints.

Our data confirmed previous evidence that the ribosome predominantly protects a 28 nt footprint in the presence of cycloheximide, and suggest that cycloheximide stabilizes one stage of the elongation cycle. Previous work shows that cycloheximide bound alongside a tRNA in the E site

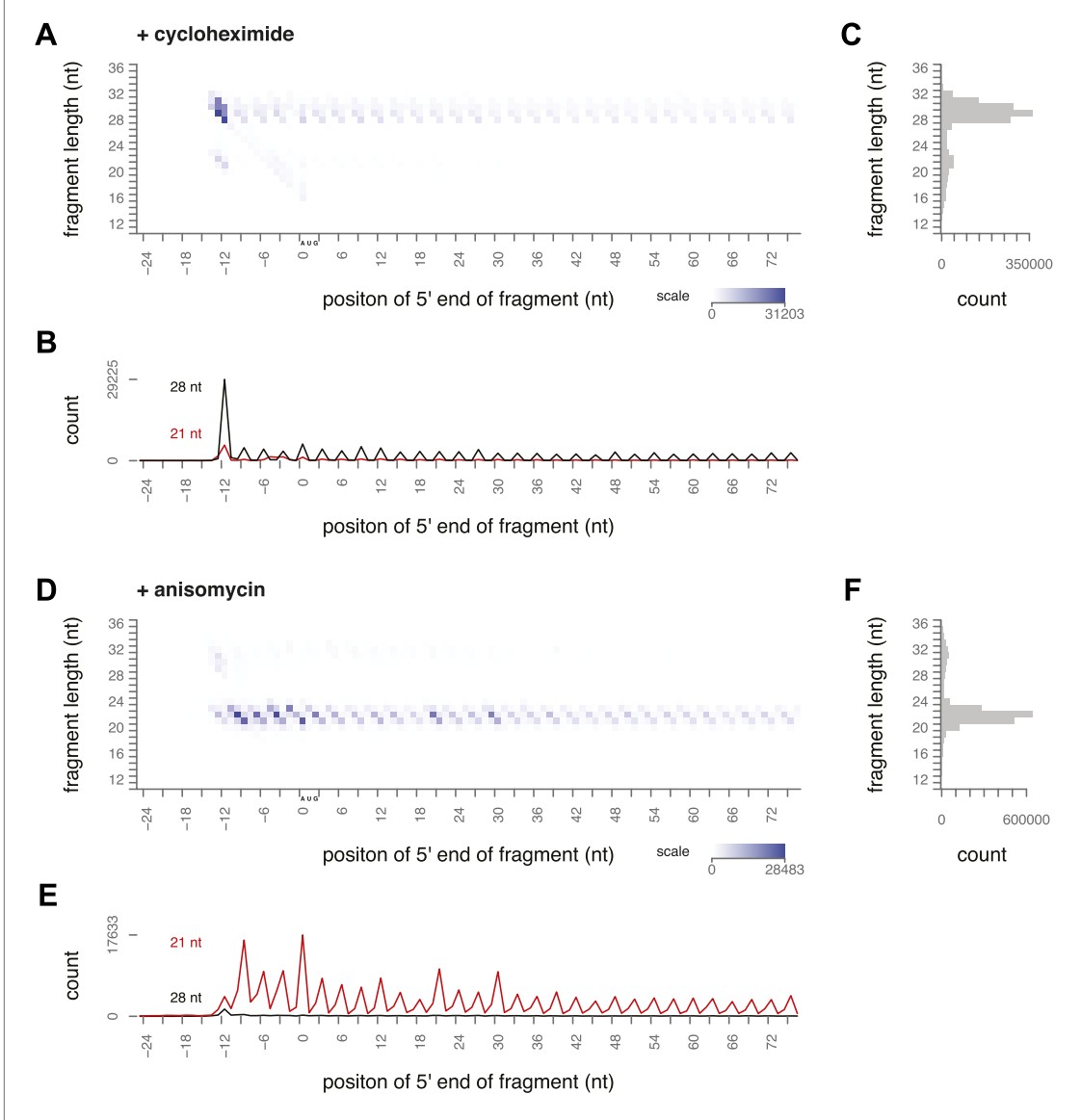

**Figure 3**. Ribosome-protected fragment positions and size distributions from yeast treated with elongation inhibitors. (**A** and **B**) As in *Figure 2A,B*, fragment position and size distribution for yeast treated with cycloheximide. (**C**) Distribution of mapped fragment lengths for yeast treated with cycloheximide. (**D** and **E**) Fragment position and size distribution for yeast treated with anisomycin. (**F**) Distribution of mapped fragment lengths for yeast treated with anisomycin.
The following figure supplements are available for figure 3:

**Figure supplement 1**. Ribosome-protected fragment positions and size distributions from yeast treated with elongation inhibitors.

prevents either the incorporation of the next aminoacylated tRNA in the A site or peptide bond formation (*Schneider-Poetsch et al., 2010*). In either case, it is expected to trap the ribosome in a non-rotated conformation, suggesting that the non-rotated conformation protects 28–30 nt of mRNA.

We next conducted ribosome-profiling experiments using yeast treated with anisomycin, an elongation inhibitor that binds to the peptidyl transferase center (*Grollman, 1967*; *Hansen et al., 2003*). We observed almost exclusively small footprints in yeast treated with anisomycin (*Figure 3D–F*, *Figure 3—figure supplement 1*). By comparison to the effects of cycloheximide treatment, we inferred that anisomycin stabilizes a distinct conformation of the ribosome that protects 20–22 nt of mRNA. Although anisomycin's precise mechanism is not characterized, it has higher affinity for post-translocation ribosomes than for pre-translocation, cycloheximide-treated ribosomes, suggesting that it preferentially binds a ribosome conformation distinct from that stabilized by cycloheximide (*Barbacid and Vazquez, 1974*, *1975*). Lincomycin and other antibiotics that bind the peptidyl transferase center induce translocation, and lincomycin-treated ribosomes prefer a rotated conformation in in vitro FRET experiments (*Fredrick and Noller, 2003*; *Ermolenko et al., 2013*). It is possible that anisomycin acts similarly to stabilize a rotated conformation.

We have thus demonstrated that two distinct ribosome conformations can be stabilized using elongation inhibitors. Stabilization of distinct conformations by two drugs resulted in a nearly complete reciprocal bias in the size of ribosome footprints, providing evidence that large and small footprints originate from distinct ribosomal conformations. We hypothesize that each ribosome cycles through both conformations, protecting first a large footprint and then a small footprint at each codon. The footprints identified by high-throughput sequencing in a ribosome-profiling experiment represent a deep sampling of ribosomes in different states, and thus the ratio of large to small footprints in untreated cells could show, at single-codon resolution, how many ribosomes are in each stage of elongation.

## Increased decoding time produces more large footprints

To enrich for ribosomes in a single, defined stage of the elongation cycle, we induced conditions expected to result in the depletion of a specific aminoacyl-tRNA and thus to increase the decoding time when the cognate codon is in the A site. We treated yeast with 3-amino-1,2,4-triazole (3-AT), an inhibitor of histidine biosynthesis, to create a specific shortage of His-acylated tRNA and cause ribosomes to pause on histidine codons (*Figure 4A*). We would therefore expect ribosomes to accumulate at histidine codons in a pre-peptide-bond conformation. Estimating codon-specific occupancy as described in more detail below, we found that the shortage of His-tRNA dramatically increased the relative abundance of large footprints from ribosomes with His codons in the A site, with minimal effect on the abundance of small footprints (*Figure 4B,C*, *Figure 4—figure supplement 1*). During the decoding phase of elongation, before peptide bond formation, the ribosome is in a non-rotated conformation (*Frank and Agrawal, 2000*; *Gao et al., 2009*); these results therefore strongly suggest that the decoding phase of elongation (the non-rotated conformation) is represented by large footprints.

## The footprint size distribution varies by codon

Recently, ribosome profiling has revealed that translation speed varies systematically by codon (*Tuller et al., 2010*; *Stadler and Fire, 2011*; *Dana and Tuller, 2012*); we hypothesized that there might be distinct codon-specific effects on the rate of the two distinct phases of elongation represented by small and large footprints.

Using data from untreated cells, we calculated the number of large and small footprints corresponding to ribosomes with a given codon in the A site, for each codon position in the yeast transcriptome. Large footprints were defined as 28 or 29 nt and small footprints were defined as 20, 21, or 22 nt with 5′ ends positioned relative to the inferred A site as depicted in *Figure 2F*. We found substantial variation in the characteristic length distribution between codons: small footprints ranged from 38 ± 12% (UUU) to 87 ± 9% (CGG) of the total footprints for a given codon identity, averaged across three replicates.

To explore this codon effect, we computed the relative occupancy of each of the 61 sense codons in the A site. We started by considering an individual gene and calculated the over- or underrepresentation of footprints at each codon position compared to the average for all codon positions in that

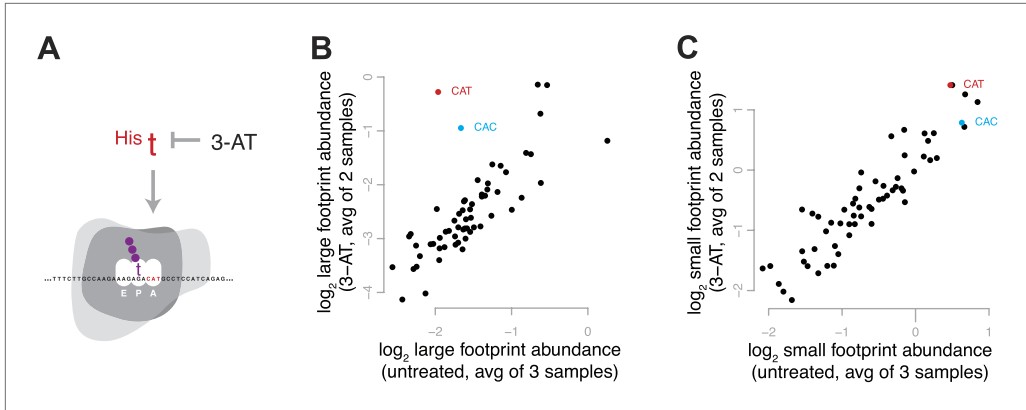

**Figure 4**. Effect of 3-amino 1,4 triazole on translation of histidine codons. (**A**) Schematic representation of the hypothesized effect of 3-AT. 3-AT reduces intracellular concentrations of histidyl-tRNA and thus is expected to increase time spent decoding histidine codons (i.e., in the decoding phase of the cycle, with a His codon in the A-site). (**B**) All 61 sense codons are plotted by the $\log_2$ of the relative abundance of large footprints with the specified codon in the A-site for untreated cells (x axis) against the $\log_2$ relative abundance of large footprints for yeast treated with 3-AT (y axis). Values shown are the average of three untreated replicates and two 3-AT treatments (10 min and 60 min). Histidine codons are denoted in red (CAT) and cyan (CAC). (**C**) As in (**B**), showing the relative abundance of small footprints.

The following figure supplements are available for figure 4:

**Figure supplement 1**. Effect of 3-amino 1,4 triazole on translation of histidine codons.

gene, including both small and large footprints (an example from a highly expressed gene is shown in *Figure 5A*). After performing this computation for every gene, we averaged these multipliers across all occurrences of a given codon in the genome to provide the 'relative occupancy' for that codon, representing, on a relative scale, how frequently we observed ribosomes with that codon positioned at the A site. The relative occupancies varied over a fivefold range, from 0.48 ± 0.04 (GGU) to 2.6 ± 0.67 (CCG) (unitless, average of three replicates) and were highly correlated between independent replicates (*Figure 5B*). As a control, we also analyzed the occupancy based on the codon one position 3′ of the A site, which has not yet entered the decoding site. We found that the range of occupancies relative to the codon in the A site was much broader than the range of occupancies relative to the next codon, suggesting that the A-site occupancies reflect an aspect of translation, not merely confounding factors such as biases in fragment capture (*Figure 5—figure supplement 1*).

Codon-specific differences in ribosome occupancy could have been driven by variation in small footprint counts, variation in large footprint counts, or both, potentially revealing the variability of each stage of elongation. We inferred the relative abundance of ribosomes in each state at each codon using a model similar to the one we used to estimate overall relative occupancy, but considering counts of either small or large footprints separately (*Figure 5A*). As with overall occupancy, the relative abundances of small footprints and the relative abundance of large footprints were both highly correlated between replicates (*Figure 5C,D*). This suggests that codon identity affected both the pre-peptide-bond and post-peptide-bond stages of elongation. However, the effect of codon identity on the inferred duration of these two phases of the elongation cycle was distinct: the codon-specific relative abundances of small and large footprints were almost uncorrelated (Spearman's r = 0.11, average of three replicates). This led us to search for physical correlates of the codon-specific differences.

## Relative occupancy is related to amino acid polarity and codon:tRNA interactions

We found that a major and unexpected determinant of the abundance of footprints from each conformation was the identity of the amino acid encoded by the A-site codon. We found a much greater density of small footprints at codons encoding smaller, polar amino acids than at codons encoding large, aromatic amino acids. The relative abundance of small footprints at codons encoding a given

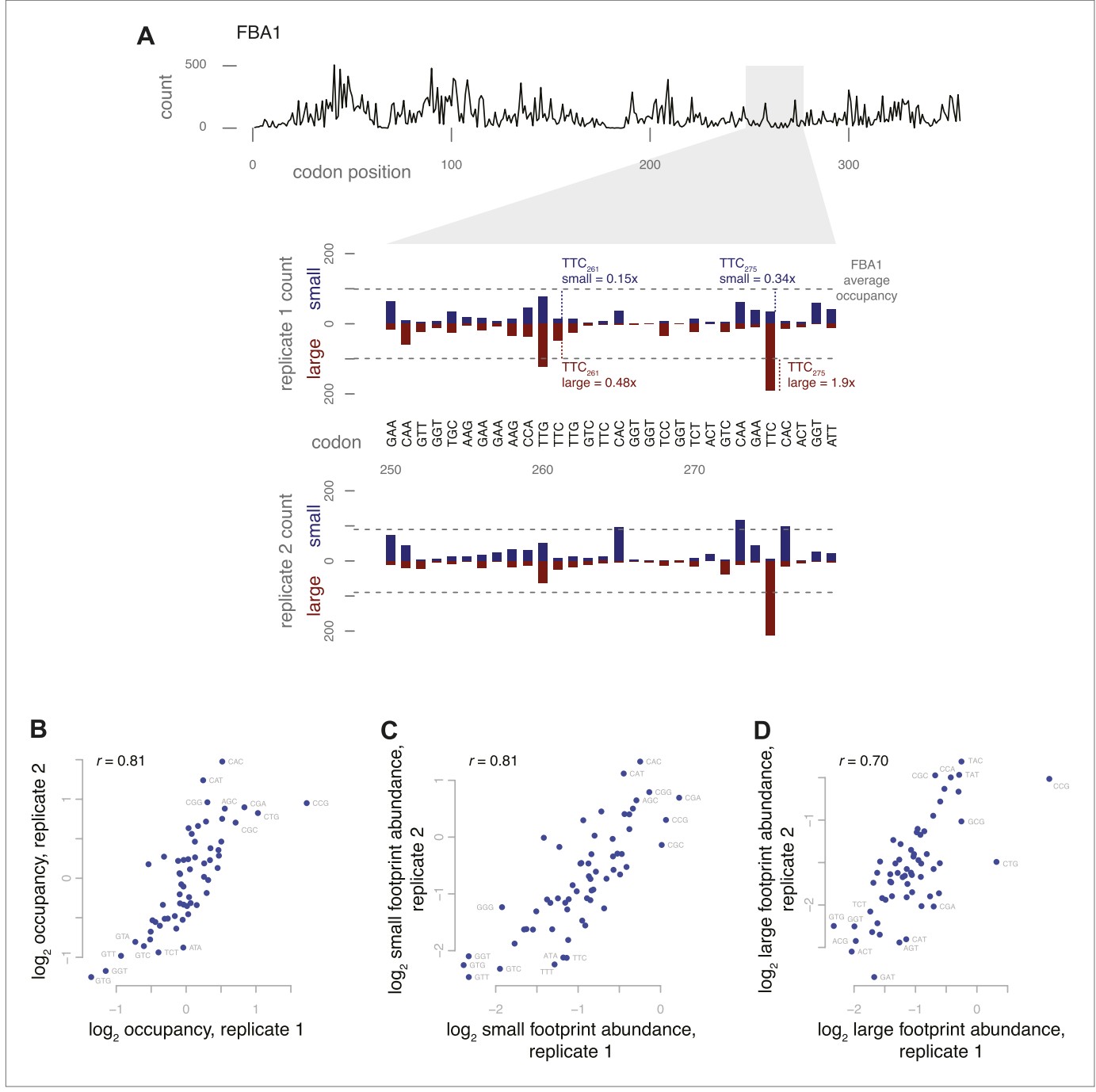

**Figure 5**. Codon-specific variation in large and small footprint abundance. (**A**) Distribution of ribosome footprint counts on the highly expressed gene FBA1, highlighting an arbitrary window, codons 250–279. Ribosome footprint counts per position were consistent between replicates and varied between instances of the same codon in this window. Relative occupancy was estimated based on the codon in the inferred A site. Total (large + small) footprint coverage at each codon of a gene was computed relative to the average coverage for that gene, then averaged by codon across all genes to provide per-codon relative occupancies. Relative abundance of small or large footprints was computed similarly, comparing the count of small or large footprints at each codon of a gene against the average coverage (large + small) for that gene, then averaged by codon across all genes. Examples of small and large footprint abundance values at two specific TTC codons in FBA1 are shown. (**B**) Relative occupancies of all 61 codons compared between two replicates, with Spearman correlation of 0.81. Stop codons and the first 50 codons of each gene were excluded from analysis. Similarly, small footprint abundance (**C**) and large footprint abundance (**D**) compared between replicates.

The following figure supplements are available for figure 5:

**Figure supplement 1**. Codon-specific variation in large and small footprint abundance.

amino acid was correlated with measures of polarity of the cognate amino acid, such as the $K_d$ of transfer of side chains from vapor to water (Spearman's r = −0.75 when grouped by amino acid, r = −0.58 by codon, *Figure 6A*), while the relative abundance of large footprints showed no correlation to amino acid polarity (Spearman's r = 0.11 by amino acid, r = 0.02 by codon) (*Wolfenden, 2007*). These data strongly suggest that the chemical properties of the amino acid specified by the codon in the A site affect the stability of the rotated, post-peptide-bond conformation of the ribosome. We hypothesize that interactions between the ribosome and polar amino acids acylated to the A-site tRNA can slow translocation substantially.

Many factors have been proposed to affect translation speed at a given codon, particularly tRNA abundance. In yeast, the number of genes encoding a specific tRNA has been shown to be highly correlated with both codon usage and cellular tRNA concentrations (*Percudani et al., 1997*). A related measure of codon optimality is the tRNA adaptation index (tAI), which attempts to rank codons in translational efficiency by accounting for tRNA copy number, wobble pairing constraints, and codon usage (*dos Reis et al., 2004*). We found that the relative occupancy per codon was only weakly correlated with tAI and with tRNA genomic copy number (Spearman's r = −0.39 and −0.28, respectively; average of three replicates) and that the tAI was not particularly correlated with the relative abundance of either small footprints or large footprints (r = −0.34 and r = −0.20, respectively; average of three replicates). Thus, unexpectedly, codon 'optimality', as represented by the tAI, does not appear to be a major determinant of relative ribosome occupancy under the conditions tested here. The 3-AT data show that in an extreme case, a limited supply of the tRNA cognate to the A-site codon slows translation during the large-footprint stage. In contrast, our overall results in untreated yeast suggest that the differences in abundance among tRNAs in wild-type cells have only a minor effect on relative ribosome occupancy of the cognate codons under optimum growth conditions.

We also investigated the relationship between wobble base pairing, relative occupancy, and the density of large and small footprints. Wobble base pairing at the A site has recently been linked with slowed elongation in humans and worms (*Stadler and Fire, 2011*). We compared codons with perfect Watson-Crick complementarity vs the synonymous codons that pair imperfectly with the same tRNA (*Johansson et al., 2008*). While we found no consistent trend toward increased occupancy at wobble-paired codons, we observed notably higher occupancy on a subset of wobble-paired codons comprising proline CCG (G-U base pairing), leucine CUG (G-U), and arginine CGA (A-I) (*Figure 6B*). For these three wobble codon outliers, we see a dramatic increase in short footprints, representing post-decoding stages of translation (*Figure 6C,D*). The arginine CGA codon is known to be a strong inhibitor of translation in yeast, and its inhibitory effect is due more to wobble decoding than tRNA abundance and may include interactions after the initial decoding (*Letzring et al., 2010*). Our data confirm that CGA is indeed one of the most slowly translated codons, and its high relative occupancy is due to increased abundance of small footprints, suggesting that its slow elongation is primarily due to a prolonged post-decoding stage. Overall, the abundance of footprints from each step of elongation was clearly affected by several distinct codon-specific features with sometimes synergistic and sometimes opposing effects.

## Discussion

A ribosome must cycle through a series of consecutive associations with mRNA to decode the message one codon at a time. The stability of the ribosome-mRNA association allows one to observe precisely where ribosomes reside on transcripts—down to the codon being decoded—by isolating and sequencing ribosome-protected mRNA fragments. We were quite surprised to discover that the ribosome protects two different footprint sizes (28–30 nt and 20–22 nt), as the original ribosome-profiling experiments and nuclease protection assays only captured the longer footprints (*Ingolia et al., 2009*). The difference is explained by the experimental conditions: the small footprints were revealed only after we left out cycloheximide, a translation inhibitor commonly used to stabilize ribosomes on mRNA for ribosome profiling. Indeed, early study of ribosome pausing found that when cycloheximide was omitted, 20–24 nt footprints accumulated in addition to the larger footprints they saw from cycloheximide-treated ribosomes (*Wolin and Walter, 1988*). As in our own experiments, the small and large footprints they observed had the same 5′ terminus and differed at the 3′ end.

We propose that the two footprints sizes originate from two ribosome conformations corresponding to different stages of elongation: large footprints from non-rotated ribosomes during the decoding stage before peptide bond formation, and small footprints from rotated ribosomes during the translocation

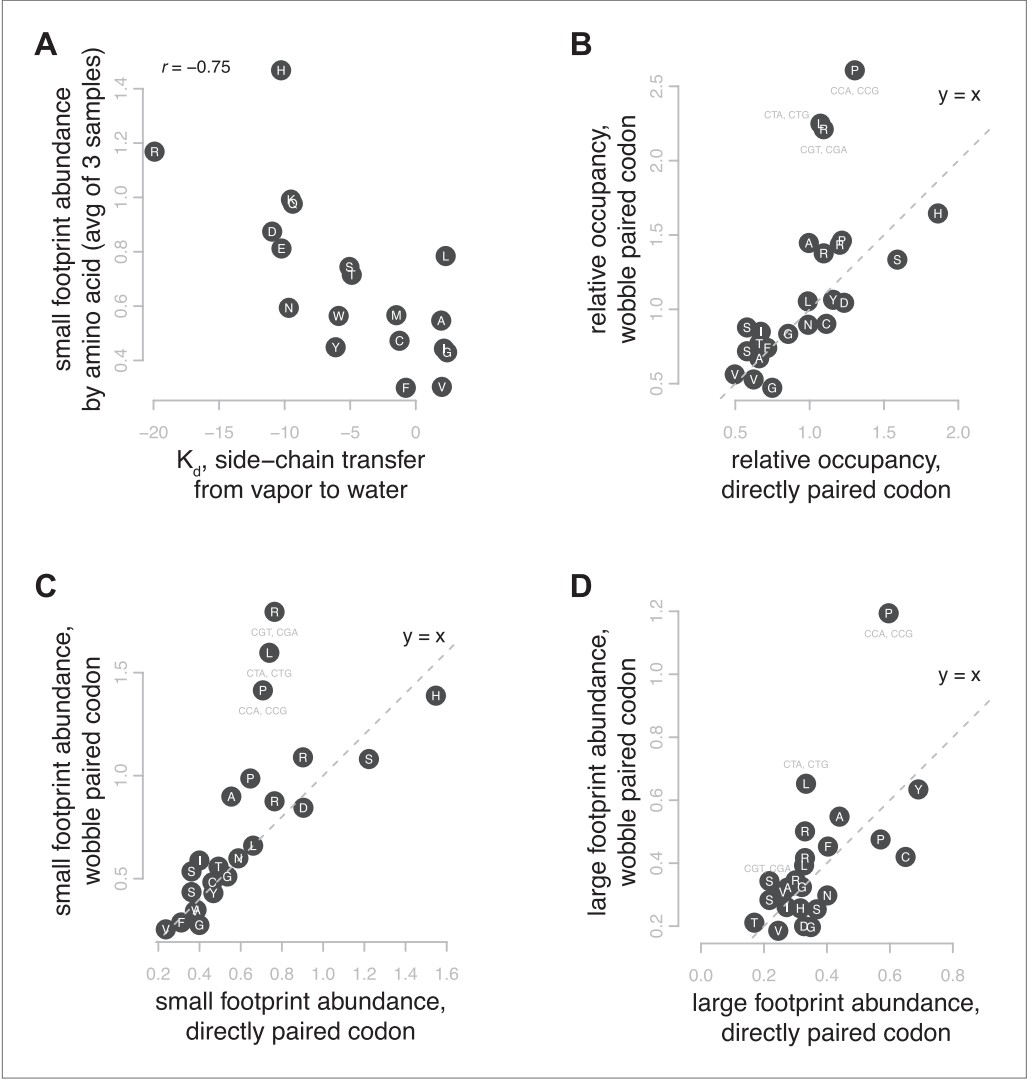

**Figure 6**. Correlates of footprint abundance. (**A**) Small footprint abundance, averaged for all codons encoding the same amino acid plotted against $K_d$ of transfer of side chain from vapor to water as a measure of polarity (**Wolfenden, 2007**), with Spearman correlation from the average of three samples. (**B**) Relative occupancy of directly paired codons vs relative occupancy of codons that recognize the same tRNA with wobble pairing. Values are the average of three replicates. Dashed line shows y = x, the expected relationship if occupancy were determined solely by tRNA identity. (**C** and **D**) As in (**B**), showing small and large footprint abundance.

stage after peptide bond formation. Additional biochemical and structural studies will be required to pinpoint the exact stages of elongation and ribosome conformations responsible for the two footprint sizes. It is not clear which of the known conformational changes during the elongation cycle are most relevant: the inter-subunit rotation after peptide bond formation, the intra-subunit swivel of the 30S head during translocation, or smaller rearrangements such as movement of the L1 stalk.

As for the physical origin of the small and large mRNA fragments, crystal structures of rotated and non-rotated ribosomes show that mRNA accessibility is not likely to be dramatically different between the two conformations (**Ben-Shem et al., 2010**, **2011**). RNAse I may be small enough to penetrate into the mRNA entrance channel and cleave the mRNA just two nucleotides from the A site. Alternately, the ribosome itself may be more susceptible to RNAse degradation in the rotated conformation, allowing ribosomal RNA cleavage that in turn enables RNAse I to access the mRNA entrance channel, yielding a smaller mRNA footprint. Importantly, however, both small and large footprints have also

been observed in wheat germ extract treated with micrococcal nuclease, indicating that the two footprint sizes are neither species- nor nuclease-specific (*Wolin and Walter, 1988*).

We hypothesize that the relative abundance of large and small footprints reflects the relative duration of different stages of elongation at each codon. (We use the A site codon by default in this discussion, though in principle we could compile results based on the codon in the P site or any other frame of reference.) Comparing our relative occupancy values to an estimated bulk elongation rate of 5.6 amino acids per second (*Ingolia et al., 2011*), our model would predict variation in average codon elongation time from as little as 0.08 s (GGU) to as much as 0.5 s (CCG). A number of caveats apply to this interpretation, and any hypotheses must be pursued with complementary approaches. Ribosome footprint data have inherent biases from ligation and other steps of the library preparation. Further, the overall balance of small and large footprints varied between replicates, leaving open the question of which conformation is more populated in vivo. Some variability arises from the mRNA fragment isolation. In this work, we chose size markers of 18 and 32 nt, but size selection from polyacrylamide gel is imprecise. (This choice also limits what we can observe: recent work found distinct 16 nt fragments from ribosomes stalled on truncated mRNAs, *Guydosh and Green, 2014*.) The size distribution may also reflect differential efficiency of library preparation from smaller or larger fragments. Nonetheless, although the overall ratio of small to large footprints varied, the codon-specific variation in this ratio was robust.

Our results also highlight the effects of harvest methods and inhibitors such as cycloheximide on footprint distribution. Ribosomes are depleted from the first 50 codons when yeast are harvested by the procedure we used without inhibitors. We interpret this as evidence that elongation continues for around 10 s after initiation ceases during the harvest process. Because the selective depletion of ribosomes from this part of the mRNA could enrich for special cases, we excluded the first 50 codons from our analysis of per-codon footprint distributions. Different harvest methods had large effects on the precise footprint locations even when the overall translation per gene was highly reproducible (data not shown). Similarly, the average occupancies per codon with and without cycloheximide were surprisingly uncorrelated (Spearman's r = 0.02, comparing the average of three untreated samples and the average of two cycloheximide-treated samples), though the total footprints per gene correlated quite well (Spearman's r = 0.97 between the average fpkm in three untreated samples and the average fpkm in two cycloheximide-treated samples). Ribosomes in different positions may be differentially affected either by the drug treatment or by runoff elongation during harvest without inhibitors. In either case, some ribosomes may halt while others undergo several more rounds of elongation.

There are many potentially rate-controlling steps of elongation and many factors necessary for each cycle, including aminoacylated tRNA and elongation factors eEF1, eEF2, and the yeast-specific eEF3 (*Kapp and Lorsch, 2004*). For example, interactions between the tRNA anticodon and the mRNA codon, the tRNAs and the ribosome, the amino acids and the peptidyl transferase center, and the nascent peptide and the tunnel, as the tRNAs move through the A, P, and E sites, can all presumably affect the speed of each step. Thus, the speed of each elongation cycle is expected to be influenced by codon, tRNA, and amino acid identity.

One of the surprising aspects of this study is that tRNA abundance or codon optimality failed to predict variation in observed ribosome occupancy and, further, that much of the variation in codon-specific occupancy was in the steps following decoding and peptide bond formation. Biochemical evidence suggests that evolution has tuned tRNA sequence and modifications to balance the contributions of amino acid identity, codon pairing strength, and tRNA structure to binding affinity of a given tRNA, such that most aminoacylated tRNAs have similar affinity to ribosomal A sites (*Olejniczak et al., 2005*; *Dale et al., 2009*; *Shepotinovskaya and Uhlenbeck, 2013*). While this affinity tuning is a plausible result of selection for fidelity in decoding, ribosome profiling has revealed a lack of uniformity both in decoding and post-decoding steps. Once the interactions that determine the codon-specific rate of decoding are decoupled and replaced by a new set of codon-specific interactions in the subsequent steps of elongation, the great diversity in physical properties of amino acids and in the intrinsic stability of the codon–anticodon interaction may lead to wide variation in the kinetics of post-decoding steps.

New methods for high-throughput measurement of translation have led to renewed interest in modeling the constraints on coding sequence and the effects of codon choice on translation efficiency (*Tuller et al., 2010*; *Plotkin and Kudla, 2011*; *Dana and Tuller, 2012*; *Charneski and Hurst, 2013*; *Shah et al., 2013*). The ability to distinguish ribosome conformations at codon resolution now allows

us to map these effects to specific phases of the elongation cycle, initiation, or termination. Future in vivo and in vitro experiments using this approach to monitor the decoding and translocation steps at each codon should provide new precision in dissecting the mechanisms by which mRNA sequence, core translation factors and regulatory factors control initiation, elongation, and termination of translation.

## Materials and methods

### Yeast strains and growth conditions

For all experiments, excluding 3-AT drug treatment experiments, BY4741 was grown overnight in YPD at 30°C; two 500 ml cultures of YPD were inoculated from the overnight culture to an $OD_{600}$ of ~0.2. For experiments involving 3-AT, S288C was grown as above in SC-His media at 30°C. Cells were then grown to mid-log phase, $OD_{600}$ ~0.6, prior to harvest. (Strain information: BY4741 derived from S288C: *MATa his3Δ1/his3Δ1 leu2Δ0/leu2Δ0 lys2Δ0/LYS2 MET15/met15Δ0 ura3Δ0/ura3Δ0. S288C: MATa SUC2 gal2 mal mel flo1 flo8-1 hap1*.)

Cells were harvested by filtration at room temperature and then quickly frozen in liquid $N_2$. Resulting cell pellets were then pulverized using a MM301 Retsch mixer mill at 30 Hz for 3 min. All chambers and tubes were pre-frozen in liquid N2 or dry ice. Approximately 400–500 μl of cold lysis buffer (20 mM Tris pH 8.0, 140 mM KCl, 1.5 mM $MgCl_2$, 1% Triton) was added to cell powder. Resulting lysates were pre-cleared by centrifugation at 2000 rpm for 5–10 min at 4°C. Lysate was transferred to a clean pre-chilled tube and further clarified by centrifugation at 20,000×*g* for 10 min at 4°C. Lysate was then stored at −80°C until RNase digestion.

For the cycloheximide experiments, cycloheximide was added to cells prior to harvest at 100 μg/ml and was also present at 100 μg/ml in the lysis buffer. For the anisomycin experiment, anisomycin was added to mid-log cells at 100 μg/ml and cells were allowed to grow for an additional 30 min prior to harvest. Anisomycin was also present at 100 μg/ml in the lysis buffer. For the 3-AT experiments, 3-amino-1,2,4-triazole was added to mid-log cells to reach a final concentration of 100 mM, then cells were grown with shaking for 10 and 60 min prior to harvest.

### RNase digestion and monosome isolation

RNase digestion and monosome isolation were performed similar to *Ingolia et al. (2009, 2012)*. Cell lysate (~800 μg total RNA measured by Nanodrop) was allowed to thaw on ice. 600 U of RNase I (AM2294; Life Technologies, Carlsbad, CA) was added to cell lysate and placed on a nutator at room temperature for 1 hr. A second cell lysate served as an undigested control; 120 U of SUPERase-In was added and placed on a nutator as above. Linear 10–50% sucrose gradients were prepared using a BioComp Gradient Master (Biocomp Instruments, Fredericton, Canada) according to manufacturer's instructions. Sucrose was dissolved in 20 mM Tris pH 8.0, 140 mM KCl, 5 mM $MgCl_2$, 0.5 mM DTT, 20 U/ml SUPERase-In; 100 μg/ml cycloheximide or 100 μg/ml anisomycin were added to buffer for corresponding experiments. After RNase digestion, lysate was added to the top of gradients and sedimented at 35,000 rpm in a SW41 rotor for 3 hr.

Gradients were fractionated at 0.17 mm per second using the BioComp Gradient Master while the $A_{260}$ was continuously monitored. Fractions corresponding to the monosome peak were collected and pooled. RNA was then purified using a miRNeasy Mini kit from Qiagen (cat# 217004; Qiagen, Venlo, Netherlands) as per manufacturer's instructions.

### Library preparation and high-throughput sequencing

Ribosome footprint libraries were prepared similar to *Ingolia et al. (2012)*. Purified RNA was separated on a 15% TBE-Urea gel. RNA oligonucleotides of 18 and 34 nucleotides were run side by side with isolated RNA and used as size markers to cut RNA of desired size for gel extraction. Size-selected RNA fragments were then treated with polynucleotide kinase to remove the 3′ phosphate. After isopropanol precipitation, dephosphorylated fragments were ligated to Universal miRNA cloning linker from New England Biolabs, Ipswich, MA (cat# S1315S). Ligated fragments were separated from excess linker by gel electrophoresis on a 15% TBE-Urea gel. After gel extraction, ligated fragments were then reverse transcribed using SuperScript III from Life Technologies (cat# 18080-085) according to manufacturer's instructions. Reverse transcriptase reactions were primed with 1 μl of 1.25 μM NI-NI-9 primer (*Supplementary file 1*). Additionally 20 U of SUPERase-In was added to each RT reaction. Reactions were incubated at 48°C for 30 min.

After reverse transcription, RNA template was removed by the addition of 2.2 μl of 1 N NaOH and incubation at 98°C for 20 min. After precipitation, cDNA was separated from excess primer by gel electrophoresis on a 5% TBE-Urea gel. cDNA was then circularized using CircLigase ssDNA ligase from Epicentre, Madison, WI (cat# CL4115K) according to manufacturer's instructions. After circularization, 5 μl of the circularization reaction was added to 1 μl of pooled ribosomal subtraction oligos (*Supplementary file 1*), 1 μl of 20x SSC, and 3 μl of water. Each sample was then denatured for 90 s at 100°C and then annealed to 37°C. MyOne Streptavidin C1 DynaBeads (25 μl per reaction) were washed three times in 1x Bind/Wash buffer (1 M NaCl, 1 mM EDTA, 10 mM Tris, pH 8.0). Beads were then resuspended in 2x Bind/Wash buffer (10 μl per reaction). Beads were added to each cDNA/oligo mixture and incubated for 15 min at 37°C in an Eppendorf ThermoMixer at 1000 rpm. Beads were collected on a magnetic stand and ~17.5 μl of eluate was recovered for each reaction. Resulting eluate was then used as a template for PCR amplification.

Pilot PCR reactions were prepared in order to determine the number of cycles necessary for adequate amplification. PCR reactions consisted of 20 μl of 5x HF buffer, 2 μl of 10 mM dNTPs, 0.5 μl of 100 μM NI-NI-2 primer, 0.5 μl of 100 μM indexing primer (*Supplementary file 1*), 5 μl of eluate template, 71 μl of water and 1 μl of Phusion polymerase (cat# M0530L; NEB). Each 100 μl reaction was separated into five 16.7 μl aliquots. PCR conditions were as follows: initial denaturation for 30 s at 98°C, followed by cycles of 10 s at 98°C, 10 s of annealing at 65°C, and 5 s of extension at 72°C. One aliquot was removed after 8, 10, 12, and 14 cycles. Amplification was examined by gel electrophoresis on an 8% TBE polyacrylamide gel. Once optimal cycle was determined, an additional 100 μl PCR was performed and run on an 8% TBE polyacrylamide gel. The product band was then cut out and DNA extracted from the gel slice. Libraries were quantified by Bioanalyzer using a DNA High Sensitivity kit (cat# 5067-4626; Agilent, Santa Clara, CA). Libraries were then sequenced on an Illumina Genome Analyzer 2 according to manufacturer's instructions by the Stanford Functional Genomics Facility. Raw sequence data are available in the Gene Expression Omnibus under accession GSE58321.

## Sequence alignment and analysis

Cloning linker sequences were trimmed from Illumina reads and the trimmed fasta sequences were aligned to *S. cerevisiae* ribosomal and noncoding RNA sequences using bowtie v. 0.12.7 or v. 1.0.0 to remove rRNA reads (*Langmead et al., 2009*). The non-rRNA reads were aligned to the *S. cerevisiae* genome as a first pass to remove any reads that mapped to multiple locations. Reads that passed this filter (those that mapped uniquely to the genome, or those that did not map at all, such as splice junction reads) were then aligned to the *S. cerevisiae* transcriptome with bowtie, allowing two mismatches and only reporting alignments of reads that mapped uniquely in the transcriptome (bowtie -v 2 -m 1 -a --norc --best –strata).

The *S. cerevisae* transcriptome sequences were based on CDS sequences downloaded from the UCSC genome browser, sacCer2 assembly, in August 2011. Untranslated region coordinates were taken from supplemental table S4 of *Nagalakshmi et al. (2008)*. When no UTR was annotated, 50 nt upstream and/or downstream of the CDS was included by default.

A list of read counts and read lengths per nucleotide position in the transcriptome, based on the 5′ end of the mapped read, was generated. From that list, metagene grids as in *Figure 2A* were made by tabulating all footprints 11–36 nt long within the following regions: last 25 nt of 5' UTR, first 200 nt of CDS, last 100 nt of CDS, and first 50 nt of 3' UTR, for all genes with a CDS of at least 300 nt.

## Per-codon analysis

Non-unique positions in the transcriptome were filtered by splitting the yeast transcriptome into all overlapping 20mers, mapping this set of all 20mers back to the transcriptome with bowtie, and collecting the mapped locations of any 20mers with more than one perfect match in the transcriptome.

The counts of small and large footprints from ribosomes with each codon positioned in the inferred A site were generated from the list of reads at each nucleotide position as depicted in *Figure 2F*. The large footprints were defined as 28 nt reads with the 5′ end 15 nt upstream of the codon at position $i$, and 29 nt reads with the 5′ end 16 nt upstream of $i$. Small footprints included 20 nt and 21 nt reads with the 5′ end 15 nt upstream of $i$ and 21 nt and 22 nt reads with the 5′ end 16 nt upstream of $i$. For each gene, the analysis included codons 51 through the second codon before the stop codon, to avoid the region at the beginning of genes from which ribosomes have been depleted by runoff elongation during harvest. Genes with fewer than 10 footprints in total were excluded, as were any non-unique positions within genes. Processed data are available in the Gene Expression Omnibus under accession GSE58321.

The 'relative occupancy' per codon was generated by first computing the average number of footprints (large + small) across the gene. Then, at each position $i$ in gene $g$, compute (large + small at position $i$)/(average large + small in gene $g$). These ratios were then averaged across all instances of a given codon (e.g., CGA) in the transcriptome to give the relative occupancy.

The densities of small and large footprints were computed as above: (small at $i$)/(average large + small in gene $g$) and similarly (large at $i$)/(average large + small in gene $g$).

## Acknowledgements

We thank Nicholas Ingolia, Harry Noller, and Jody Puglisi for helpful discussion.

## Additional information

### Funding

| Funder | Grant reference number | Author |
| --- | --- | --- |
| Howard Hughes Medical Institute | | Liana F Lareau, Dustin H Hite, Gregory J Hogan, Patrick O Brown |
| Damon Runyon Cancer Research Foundation | DRG-2033-09 | Liana F Lareau |
| National Science Foundation | | Dustin H Hite |

The funders had no role in study design, data collection and interpretation, or the decision to submit the work for publication.

### Author contributions

LFL, DHH, Conception and design, Acquisition of data, Analysis and interpretation of data, Drafting or revising the article; GJH, Acquisition of data, Drafting or revising the article; POB, Conception and design, Analysis and interpretation of data, Drafting or revising the article

## Additional files

### Supplementary file

• Supplementary file 1. Primer sequences.

### Major dataset

The following dataset was generated:

| Author(s) | Year | Dataset title | Dataset ID and/or URL | Database, license, and accessibility information |
| --- | --- | --- | --- | --- |
| Lareau LF, Hite DH, Hogan GJ, Brown PO | 2013 | Distinct stages of the translation elongation cycle revealed by sequencing ribosome-protected mRNA fragments | GSE58321; http://www.ncbi.nlm.nih.gov/geo/query/acc.cgi?acc=GSE58321 | Publicly available at the NCBI Gene Expression Omnibus (http://www.ncbi.nlm.nih.gov/geo/). |

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
