## [Decision Letter]

Thank you for sending your work entitled “Distinct stages of the translation elongation cycle revealed by sequencing ribosome-protected mRNA fragments” for consideration at *eLife*. Your article has been favorably evaluated by a Senior editor and 3 reviewers, one of whom, Roy Parker, is a member of our Board of Reviewing Editors.

The Reviewing editor and the other reviewers discussed their comments before we reached this decision, and the Reviewing editor has assembled the following comments to help you prepare a revised submission.

Major issues:

1) A critical aspect of the work is that the 28 and 20-21 nucleotide fragments have the same 5' end and therefore can be mapped to similar codons. The data presented in Figure 2 are consistent with an interpretation that the short fragments are truncated at their 3' (rather than their 5') ends. That, taken together with the antibiotic and 3-AT data, makes for a strong overall conclusion. However, the figures are extremely confusing and fail to clearly convey this interpretation. While the 3D presentation of data in 2A is helpful, the authors failed to include a z-axis scale to calibrate the varied levels of shading. Also, they should have “projected” the data from short and long fragment distributions onto two more standard “metagene” style plots above of graph (much like 2C serves as a horizontal projection) since that is what most people have become familiar with in understanding this sort of data.

While the data are convincing, certain aspects of it do raise questions about the potential for random RNase degradation. For example, why do all reads in between the 28- and 20-nt peaks seem to derive from a region around the start codon? In the plot, this appears as a diagonal line extending from (-12,28) on the plot to (0,15). Perhaps this could be addressed? Ideally, the authors would have done some kind of an RNase protection assay in vitro, possibly with multiple nucleases, to establish the certainty of their results. In the end, the authors are arguing that the alternative conformation of the ribosome is hyper-susceptible to RNase cleavage, thus revealing the mRNA to the nucleases in a way that is surprising given what we know about ribosome structure. So the clearer this section is the better. In addition, perhaps it would help to emphasize earlier and more emphatically that Walter et al. saw these fragments long ago in his in vitro experiments.

In a related issue, the overall pattern of 20 nt reads in Figure 2—figure supplement 1 had huge variation at each position and did not match the 28 nt reads pattern from the same sample. These results do not support that both populations of RPF reads have the same 5' end. This needs to be clarified.

2) A second key issue is the reproducibility of the data presented. It is unclear from the text whether these experiments have been done more than once in each case. This is an issue since the authors’ present data that differences in harvesting protocols can affect the observed data. Given this, it should either a) be clarified that different experiments have been done more than once with good reproducibility, or b) the experiments repeated to demonstrate reproducibility.

3) The discussion/presentation of data on start and stop codon should be improved.

In general, wide variation in relative ribosome occupancies have been reported at start and stop codons. For example, it has been suggested that CHX treatment leads to high occupancy at start codons as initiation of ribosomes continues despite the inability of the ribosome to move away from the start codon (Ingolia et al., Nature Protocols 2012). Therefore, the authors should be more cautious in interpreting a selective enrichment of small fragments at start codons in CHX-treated yeast. At the least, the authors should have performed replicate experiments to verify that slight variation in sample preparation isn't leading to this effect. Similarly, the effects at stop codons, while intriguing, ought to be taken with a grain of salt because of widespread variation in the level of previously reported stop codon occupancy. Unlike the data reported here, higher than average occupancy at stop codons was shown in wild-type cells in Ingolia et al. Cell 2011, Liu et al. Cell 2013, and Lee et al. PNAS 2012. The authors' own experiments seem to suggest that this variability may be sometimes derived from differences in preparation (they show that variation in cell freezing rate can dramatically affect occupancy at stop codons). For these reasons, additional replicates are needed (and should be explicitly documented) to make these interpretations believable.

The results specific to the dom34 knockout yeast, while again intriguing, should not be overinterpreted. How many times were these effects observed?

Other factors, such as shifts in the placement of the footprint are not considered in this section. Such shifts have been reported by Tatyana Pestova and should be addressed and cited (Alkalaeva et al., Cell 2006) as they may explain why occupancy isn't always seen exactly 15 nt upstream at the termination codon.

Finally, some of the conclusions from these sections are not particularly insightful and often overinterpreted. For example, “dom34 has an important general role in ribosome recycling” is overly generic. Similarly, “following formation of this 'nonsense-suppressor' peptide bond, the ensuing translocation step appears to be dramatically slowed” is purely speculative. How do we know that the high-occupancy footprints correspond to ribosomes that are decoding the stop codon vs. stalled with eRF1 bound?

In general, this section should be trimmed back as it seems superficial to the main claims presented earlier. These results (specifically all the data in Figure 6) together distract from the already very interesting story presented earlier in the paper and should be left for further development in a subsequent study. If the authors insist on inclusion, then replicates on each sample must be performed and documented.

4) The authors should avoid improper terms about time-dependent effects. The authors have made generous use of terms like “speed,” “rate,” and “dwell time” throughout the manuscript when referring to their data. Such language is not justified because no time course experiments were performed here. While it is conceivable (and even likely) that ribosome occupancy levels may correspond to actual ribosome dwell times, this has never been proven. We do not know, for example, that either of the footprint sizes studied here are derived from a ribosome in a rate-limiting waiting state during peptide elongation. Comparisons between rate and occupancy should be left to the Discussion section, where such speculative analysis is more appropriate. Moreover, it is unclear to discuss data with these terms or show figures with axes labeled with “dwell time” (Figures 4 and 5) or “speed” (Figure 5) because these quantities were not measured (the units given are not seconds or inverse seconds) and they burden the reader to understand intricate models for how these quantities were derived. The variable of interest here is “ribosome occupancy” or “fraction of reads” and must be used when describing the data for the sake of both clarity and honesty.

5) Several areas of the manuscript employ language that is unclear or suggests ideas that are not necessarily true and should be fixed.

i) In the abstract, the ribosome is said to “ratchet” along the mRNA. “Ratchet” is a fairly loaded word that signifies certain types of biophysical model and should be replaced by something less specific.

ii) It was unclear in the main text whether CHX was added to cell culture or simply used in the lysis buffer, since both approaches have been reported previously. This should be explicit here.

iii) The language about antibiotics is also unclear. For example, the phrase “anisomycin may act like lincomycin” is confusing because nowhere is it explained how lincomycin works. Overall, it would be helpful if the authors review more of the literature on the mechanism of anisomycin and other antibiotics. Alternatively, given the often pleiotropic or poorly defined effects of antibiotics on the translation cycle (e.g., paromomycin on decoding and recycling), some caution in interpretation would be warranted.

iv) It is not clear what a “translocation conformation” signifies?; The state just before translocation? Similarly, “translocation of the second codon” is an awkward and nonstandard expression.

v) The authors state that, “the extent of ribosome pausing after 3-AT treatment may have been attenuated by the general amino acid control mechanism...” This statement makes no sense: how does alteration of the initiation rate (via the GCN2 pathway) affect the time the ribosome spends at a particular downstream codon? These would seem to be physically separate events.

vi) The authors state, “Dom34 is implicated in triggering cleavage of mRNAs.” Recent work from both Toshi Inada and Roy Parker has shown this to be untrue. Rather, Dom34 is known to rescue ribosomes on cleaved mRNAs. In general, the discussion of termination and recycling is rather wanting. Statements such as “the identity of general recycling factors in eukaryotes has been unclear” are misleading because Tatyana Pestova and Rachel Green have both implicated eRF1 and Rli1 as having specific roles in recycling at stop codons.

vii) T phrase, “evidence that ribosome profiling without inhibitors provides an accurate in vivo snapshot of translation” is unclear and likely overstated. Each profiling procedure is attempting to trap an “accurate in vivo snapshot” and I don't think we yet know which does the best job.

6) The manuscript would be improved by error analysis. The average occupancy values in Figures 4 and 5 do not contain any assessment of uncertainty for each measurement. While the data are convincing without such a statistical workup, it would be reassuring to see an example where the variation in occupancy at a particular codon is computed. In other words, what is the inherent variability in such measurements that derive from all sources of variability (variation between cultures, sequence context effects on translation rate, library preparation, etc).

7) A number of problems arise in the analysis of wobble-paired codons. Figure 5 shows the majority of codons fall on the y=x line. Thus, the authors' interpretation of these data as showing that “elongation was dramatically slower at codons with wobble pairing” seems vastly overstated. Instead, it appears that a handful of codons seem to have increased ribosome occupancy, and may therefore indicate slower translation. The authors should indicate that it may not be wobble pairing per se that is the cause here or that it may only be one particular type of wobble pairing that is problematic.

In addition, the authors appear to include more than simple wobble pairings in Figure 5. They show 3 (or possibly 4 as the labels are very difficult to read and ought to be larger and show the full codon sequence) proline codons on the plot. There are at most 2 wobble-paired proline codons given that there are 2 proline tRNAs (UGG and IGG anticodons). The UGG anticodon decodes CCA with Watson-Crick pairing and CCG with U-G wobble pairing. The IGG anticodon decodes CCU with I-U wobble pairing and CCC with an I-C, essentially Watson-Crick pairing (not the same as a wobble pairing). Overall, it would be useful if the authors distinguished between these various non-canonical pairings by giving the occupancies for each codon separately, as it may explain why only a small fraction of the codons show any adverse effects on the inferred translation rate.

8) Biases introduced during library creation.

One of the unsaid truths about high-throughput methods is that the level of amplification of a given nucleic acid sequence during library preparation is highly dependent on the content of the sequence due to bias at each of the enzymatic steps in the process (ligations, reverse transcription, PCR, etc.). The authors did not address how these biases apply to the two different sizes of ribosome-protected fragments. Are the 20mers more abundant than 28mers because they are more efficiently turned into cDNA? Does codon identity introduce different bias to 20mer preparation vs 28mer preparation? While the systematic effect of amino acid polarity for various codons would suggest biases are not important here, it would be ideal if the authors had prepared mRNA-Seq libraries that consisted of fragments of 20- and 28-nt in size to get a handle on the bias. If the authors have such data, it ought to be included in the manuscript to put this concern to rest.

---

## [Author Response]

*1) A critical aspect of the work is that the 28 and 20-21 nucleotide fragments have the same 5' end and therefore can be mapped to similar codons*. *[…]*

*While the data are convincing, certain aspects of it do raise questions about the potential for random RNase degradation. For example, why do all reads in between the 28- and 20-nt peaks seem to derive from a region around the start codon? In the plot, this appears as a diagonal line extending from (-12,28) on the plot to (0,15). Perhaps this could be addressed? Ideally, the authors would have done some kind of an RNase protection assay in vitro, possibly with multiple nucleases, to establish the certainty of their results. In the end, the authors are arguing that the alternative conformation of the ribosome is hyper-susceptible to RNase cleavage, thus revealing the mRNA to the nucleases in a way that is surprising given what we know about ribosome structure. So the clearer this section is the better. In addition, perhaps it would help to emphasize earlier and more emphatically that Walter et al. saw these fragments long ago in his in vitro experiments*.

*In a related issue, the overall pattern of 20 nt reads in*
Figure 2—figure supplement 1
*had huge variation at each position and did not match the 28 nt reads pattern from the same sample. These results do not support that both populations of RPF reads have the same 5' end. This needs to be clarified*.

We’ve added the z-axis scale and projected metagenes to Figures 2 and 3. We hope this makes our fairly dense figures easier to interpret. We performed replicate experiments as detailed below in point 2, and these replicates further support our conclusion that the short and long fragments have the same 5' end (see metagenes of the new replicates, Figure 2 and Figure 2—figure supplement 1). The diagonal line extending from (-12,28) on the plot to (0,15) is not present in all replicates. We think it is 5’ degradation of extremely abundant start codon footprints (perhaps the ribosome at the start codon is somehow more susceptible to degradation). However, we’ve removed any discussion of start and stop codons from our paper, so we haven’t investigated this further though we agree it is curious. All of our codon analyses start at codon 50 to avoid runoff (see below) and start codon artifacts at the beginning of the gene.

The supplement to Figure 2 has been removed because it was confusing and tangential. It showed the count per *codon* (summed across 3 nt using the same A-site inference rules we use later) rather than the count per nucleotide. Relative to the total metagene plots, this plot displayed higher variability since it was limited to long genes and thus included less data. Our intention was to show that one artifact of inhibitor-free profiling is runoff of ribosomes near the beginning of the gene. These data convinced us to exclude the first 50 codons from the codon-specific analysis. The metagenes, on the other hand, are a zoomed-in view of the per-nucleotide counts in the first 36 codons, a region that is relatively depleted of ribosomes – but it is essential to include the beginning of the metagene to show that the 5’ ends of the first peaks of 21 and 28 nt fragments are in the same place.

*2) A second key issue is the reproducibility of the data presented. It is unclear from the text whether these experiments have been done more than once in each case. This is an issue since the authors’ present data that differences in harvesting protocols can affect the observed data. Given this, it should either a) be clarified that different experiments have been done more than once with good reproducibility, or b) the experiments repeated to demonstrate reproducibility*.

In our original paper, we included results from one experiment under each condition. While we had previously done each of these experiments a few times with the same overall results, the data were not directly comparable as we were optimizing drug treatment and nuclease conditions.

We have now performed two more biological replicates on untreated yeast, one cycloheximide replicate, and one anisomycin replicate (which were then split into two technical replicates for all stages after we collected the yeast lysate). Along with the new replicates, we’re including one of these older samples (treatment with 3-AT for 10 minutes, along with the 60 minute treatment included in our previous manuscript) to strengthen the 3-AT results.

In our manuscript, we have clarified that the experiments have been performed multiple times. We now use the highest coverage samples for our main metagene figures and include the other replicates as supplements to Figures 2 and 3. We use the average of three replicates for codon-specific values.

We note here, since it’s a change from the original paper, that the overall ratio of small and large footprints is somewhat variable between our untreated replicates. All of the untreated samples have roughly a 50-50 mix of small & large footprints, and one state is not consistently preferred over the other. We can no longer conclude that the variation in small footprints (post-decoding) drives total occupancy, because those correlations are necessarily dependent on the overall ratio of small to large footprints. Instead, we now include much stronger evidence that both small and large footprints show variation between codons that is very consistent between replicates (Figure 5), and that these two values (abundance of large and small footprints per codon) are independent of each other. Also, we can say with more certainty that features such as amino acid polarity is correlated with the small footprint abundance (post-decoding).

Another minor change throughout the paper: because of slight differences in digested footprint size, we’ve realized that 21 nt footprints are a better representative of the small (20-22nt) population. In the few cases where we need to look at a single protected-fragment size (as in the projected standard metagenes) we have now used the 21 nt fragments. For all codon analyses we use the same 20-22 and 28-29 range as in the first version.

*3) The discussion/presentation of data on start and stop codon should be improved*.

*[…] Unlike the data reported here, higher than average occupancy at stop codons was shown in wild-type cells in Ingolia et al. Cell 2011, Liu et al. Cell 2013, and Lee et al. PNAS 2012. The authors' own experiments seem to suggest that this variability may be sometimes derived from differences in preparation (they show that variation in cell freezing rate can dramatically affect occupancy at stop codons). For these reasons, additional replicates are needed (and should be explicitly documented) to make these interpretations believable*.

*The results specific to the dom34 knockout yeast, while again intriguing, should not be overinterpreted*. *How many times were these effects observed?*

*Other factors, such as shifts in the placement of the footprint are not considered in this section. Such shifts have been reported by Tatyana Pestova and should be addressed and cited (Alkalaeva et al., Cell 2006) as they may explain why occupancy isn't always seen exactly 15 nt upstream at the termination codon*.

*[…] In general, this section should be trimmed back as it seems superficial to the main claims presented earlier. These results (specifically all the data in*
Figure 6*) together distract from the already very interesting story presented earlier in the paper and should be left for further development in a subsequent study. If the authors insist on inclusion, then replicates on each sample must be performed and documented*.

We’ve entirely removed the start and stop codon and dom34 sections from this manuscript (and hope to expand them with more evidence in a future paper). We agree with the reviewers that there is a lot of variation in stop codon occupancy between experiments; this is something we intend to explore further.

*4) The authors should avoid improper terms about time-dependent effects. The authors have made generous use of terms like “speed,” “rate,” and “dwell time” throughout the manuscript when referring to their data. Such language is not justified because no time course experiments were performed here. […] The variable of interest here is “ribosome occupancy” or “fraction of reads” and must be used when describing the data for the sake of both clarity and honesty*.

We replaced these terms with “relative occupancy” for the overall density of ribosomes on a codon and “relative abundance” for the density of small and large footprints. We think this makes our results clearer and more direct.

*5) Several areas of the manuscript employ language that is unclear or suggests ideas that are not necessarily true and should be fixed*.

*i) In the abstract, the ribosome is said to “ratchet” along the mRNA. “Ratchet” is a fairly loaded word that signifies certain types of biophysical model and should be replaced by something less specific*.

We would very much like to keep this metaphor because it evokes the motion we are studying. A number of papers describe the ribosome as ratcheting, for example, “Structures of the ribosome in intermediate states of ratcheting,” Zhang et al, Science 2009. (Some also propose the idea of the ribosome as a Brownian ratchet – a much more specific claim that we are not trying to make.) We realize that the exact nature of the pawl of the ribosome’s ratcheting motion is not known conclusively.

We know that this is an active topic of research, and if absolutely necessary we could replace “ratchets along” with “steps along.”

*ii) It was unclear in the main text whether CHX was added to cell culture or simply used in the lysis buffer, since both approaches have been reported previously. This should be explicit here*.

CHX was added to the yeast media ∼30-60 seconds before harvesting and was also present in the lysis buffer. We’ve clarified this in the main text.

*iii) The language about antibiotics is also unclear. For example, the phrase “anisomycin may act like lincomycin” is confusing because nowhere is it explained how lincomycin works. Overall, it would be helpful if the authors review more of the literature on the mechanism of anisomycin and other antibiotics. Alternatively, given the often pleiotropic or poorly defined effects of antibiotics on the translation cycle (e.g., paromomycin on decoding and recycling), some caution in interpretation would be warranted*.

Unfortunately, anisomycin is much less studied than other peptidyl transferase inhibitors. We’ve revised this section to be more careful and complete about what is known about this drug. (In retrospect we could have tried other antibiotics instead.)

*iv) It is not clear what a “translocation conformation” signifies?; The state just before translocation? Similarly, “translocation of the second codon” is an awkward and nonstandard expression*.

We rewrote the first of these (in the paragraph about anisomycin’s mechanism) and the second was in a section that we have removed entirely.

*v) The authors state that, “the extent of ribosome pausing after 3-AT treatment may have been attenuated by the general amino acid control mechanism...” This statement makes no sense: how does alteration of the initiation rate (via the GCN2 pathway) affect the time the ribosome spends at a particular downstream codon? These would seem to be physically separate events*.

We have removed the statement, but if the cells respond to insufficient histidine by reducing initiation, this mitigates the histidine deficiency, by allowing the impaired histidine biosynthetic pathway to better keep up with the (reduced) demands of protein synthesis (and thus reduced histidyl-tRNA turnover).

*vi) The authors state, “Dom34 is implicated in triggering cleavage of mRNAs.” Recent work from both Toshi Inada and Roy Parker has shown this to be untrue. Rather, Dom34 is known to rescue ribosomes on cleaved mRNAs. In general, the discussion of termination and recycling is rather wanting. Statements such as “the identity of general recycling factors in eukaryotes has been unclear” are misleading because Tatyana Pestova and Rachel Green have both implicated eRF1 and Rli1 as having specific roles in recycling at stop codons*.

We removed this whole set of results.

*vii) T phrase, “evidence that ribosome profiling without inhibitors provides an accurate in vivo snapshot of translation” is unclear and likely overstated. Each profiling procedure is attempting to trap an “accurate in vivo snapshot” and I don't think we yet know which does the best job*.

We have removed the statement.

*6) The manuscript would be improved by error analysis. The average occupancy values in*
Figures 4 and 5
*do not contain any assessment of uncertainty for each measurement. While the data are convincing without such a statistical workup, it would be reassuring to see an example where the variation in occupancy at a particular codon is computed. In other words, what is the inherent variability in such measurements that derive from all sources of variability (variation between cultures, sequence context effects on translation rate, library preparation, etc)*.

We’ve replaced the individual values in most figures with the average of three replicates, which reduces noise and spurious associations. We also compare the values between replicates in the new Figure 5. We’ve included an arbitrary example of footprint distribution along FBA1 (one of the few genes with high enough coverage) to give a qualitative picture of the variability from position to position and the surprising consistency between replicates.

We’ve chosen not to include explicit error measurements for a few reasons. First, count-based RNA sequencing is not modeled properly by conventional error models (read counts are overdispersed and not normally distributed). Second, the coverage in our samples (even the new replicates) is low enough that many genes only have a small handful of footprints, so we’d be comparing a lot of sites with few or no reads (i.e., the variability is not informative). In the future we hope to scale up to much higher coverage to allow position-by-position analysis instead of averages.

*7) A number of problems arise in the analysis of wobble-paired codons.*
Figure 5
*shows the majority of codons fall on the y=x line. Thus, the authors' interpretation of these data as showing that “elongation was dramatically slower at codons with wobble pairing” seems vastly overstated. Instead, it appears that a handful of codons seem to have increased ribosome occupancy, and may therefore indicate slower translation. The authors should indicate that it may not be wobble pairing per se that is the cause here or that it may only be one particular type of wobble pairing that is problematic*.

*In addition, the authors appear to include more than simple wobble pairings in*
Figure 5*. They show 3 (or possibly 4 as the labels are very difficult to read and ought to be larger and show the full codon sequence) proline codons on the plot. […] Overall, it would be useful if the authors distinguished between these various non-canonical pairings by giving the occupancies for each codon separately, as it may explain why only a small fraction of the codons show any adverse effects on the inferred translation rate*.

We’ve reworded this section to make it clearer (and remade the associated figure incorporating data from replicates). For our pairing rules we had used data from Johansson et al that found that certain proline and leucine tRNAs could bind three or four codons, not two. We’ve updated our pairing rules to remove those nonstandard pairings and we’ve labeled the outlying points with the codon sequences.

*8) Biases introduced during library creation*.

*One of the unsaid truths about high-throughput methods is that the level of amplification of a given nucleic acid sequence during library preparation is highly dependent on the content of the sequence due to bias at each of the enzymatic steps in the process (ligations, reverse transcription, PCR, etc.). The authors did not address how these biases apply to the two different sizes of ribosome-protected fragments. Are the 20mers more abundant than 28mers because they are more efficiently turned into cDNA? Does codon identity introduce different bias to 20mer preparation vs 28mer preparation? While the systematic effect of amino acid polarity for various codons would suggest biases are not important here, it would be ideal if the authors had prepared mRNA-Seq libraries that consisted of fragments of 20- and 28-nt in size to get a handle on the bias. If the authors have such data, it ought to be included in the manuscript to put this concern to rest*.

This is an interesting question and we’ve added Figure 5—figure supplement 1 and a few lines to the Results and Discussion to address it. The supplementary figure compares occupancies relative to the A site codon and, as a control, occupancies relative to the next codon past the A site. We find that the not-yet-translated position has much less variance in occupancy, which is what we’d expect as this position should not be affecting translation substantially. If library creation bias had a big effect on which sequences (of either size) were captured, we would see similar ranges of variability in both.

We’ve also done a preliminary analysis of biases in the ends of footprint sequences and found no strong bias affecting which footprints sequences are captured by the method (not included in the manuscript). We don’t know if 20 or 28 nt fragments overall are captured more efficiently. In light of the variability in overall small footprint abundance between replicates, we now look at variation between different codons within each experiment separately.